# SEARCHFIRESAFETY:
# A RETRIEVAL-AUGMENTED LEGAL QA DATASET
# FOR FIRE SAFETY

## ABSTRACT

Retrieval-augmented generation (RAG) promises to bridge complex legal statutes and public understanding, yet hallucination remains a critical barrier in real-world use. Because statutes evolve and provisions frequently cross-reference, maintaining *temporal currency* and *citation awareness* is essential, favoring up-to-date sources over static parametric memory. To study these issues, we focus on the under-examined domain of South Korean fire safety regulation—a complex web of fragmented legislation, dense cross-references, and vague decrees. We introduce SEARCHFIRESAFETY, the first RAG-oriented question-answering (QA) resource for this domain. It includes: (i) 941 real-world, open-ended QA pairs from public inquiries (2023–2025); (ii) a corpus of 4,437 legal documents from 117 statutes with a citation graph; and (iii) synthetic single-hop (Yes/No) and multi-hop (MCQA) benchmarks targeting legal reasoning and uncertainty.

Experiments with five Korean-capable LLMs show that: (1) multilingual dense retrievers excel due to the domain's mix of Korean, English loanwords, and Sino-Korean terms (i.e., Chinese characters); (2) grounding LLMs with SEARCHFIRESAFETY substantially improves factual accuracy; but (3) multi-hop reasoning still fails to resolve conflicting provisions or recognize informational gaps. Additionally, we find that (4) domain adaptation via continued pre-training improves accuracy but significantly degrades uncertainty awareness when evidence in insufficient. Our results affirm that RAG is necessary but not yet sufficient for legal QA, and we offer SEARCHFIRESAFETY as a rigorous testbed to drive progress in Legal AI. All data resources are available at: https://anonymous.4open.science/r/SearchFireSafety-C2AB/.

## 1 INTRODUCTION

Retrieval-Augmented Generation (RAG) (Lewis et al., 2020) helps bridge the gap between complex technical information and public understanding. Recent work demonstrates its promise in medicine (Zakka et al., 2024), climate science (Biswas et al., 2025), and finance (Iaroshev et al., 2024; Choi et al., 2025). However, research on RAG has not fully resolved issues of inconsistency and hallucination, often triggered by noisy or irrelevant retrieved documents (Shuster et al., 2021; Chen et al., 2024b). The risk of hallucination is a critical bottleneck in safety-sensitive domains such as law and regulation, where potential impact is high and the risks require careful mitigation (Magesh et al., 2024).

In this work, we study RAG in the under-examined but socially important domain of South Korean fire-safety law. Fire-safety compliance directly affects everyday stakeholders—building owners, school administrators, small businesses, and local officials—who routinely consult regulations to determine eligibility, required installations, and responsibility. However, the legal framework governing fire safety is complex and frag-

**SearchFireSafety**

Figure 1: Overview of the proposed framework and datasets. (1) Collection of real-world QA pairs from the Korean National Fire Agency petition portal. (2) Construction of a temporally current legal corpus with human-in-the-loop remediation of non-text artifacts and a hyperlink-induced citation graph. (3) Generation of synthetic QA to evaluate hallucination in the legal domain.

mented, encompassing the *Building Act*, the *Framework Act on Fire Services*, the *Act on the Installation and Management of Fire-Fighting Systems*, and the *Special Act on the Safety Control of Publicly Used Establishments*, among others (Kodur et al., 2020; Song, 2023).

South Korean fire-safety law also poses distinctive challenges for RAG evaluation beyond language alone. The application of RAG in the legal domain presents challenges that are significantly more pronounced than in general-purpose settings due to two primary factors. First, retrieval is hindered by a significant semantic gap between informal user queries and formal legal terminology, which poses a particular problem for sparse retrieval methods. Second, legal documents are interconnected through a dense web of statutory cross-references and hierarchical delegations via presidential decrees and administrative rules, many of which are vague or overly broad (Song, 2023; Cho & Kim, 2024). For instance, Article 7 of the Enforcement Decree of the Fire-Fighting Act references over a dozen statutes, including the *Building Act*, *Child Welfare Act*, and *Mental Health Act*, creating a dense and intricate citation graph that is difficult for non-experts to navigate.

To study these challenges, we introduce SEARCHFIRESAFETY, the first question answering (QA) dataset tailored to Korea's fire safety legal domain. We collect 941 real-world, open-ended QA pairs and ground them in a corpus of 4,437 legal documents. We also construct a legal citation graph to map interconnections within the corpus. Based on this graph, we generate synthetic legal reasoning questions that mirror the domain's complexity and robustly evaluate agent performance, especially under retrieval failures.

Using SEARCHFIRESAFETY, we evaluate five Korean-capable Large Language Models (LLMs) across diverse RAG strategies. Our experiments demonstrate that grounding models in our structured dataset substantially improves factuality and alignment. Furthermore, our inclusion of synthetic datasets enables an evaluation of model performance under retrieval failure, specifically testing the models' reasoning capabilities and uncertainty awareness. We also explore domain adaptation via continued pretraining (CPT), revealing a critical trade-off: while CPT enhances accuracy with complete information, it significantly impairs a model's ability to abstain when information is missing.

In summary, our main contributions are as follows:

• We introduce SEARCHFIRESAFETY, the first dedicated QA benchmark for the South Korean fire-safety legal domain. By constructing a legal citation graph, we also generate synthetic multi-hop reasoning questions to rigorously test model performance in complex regulatory environments.
• We conduct a comprehensive evaluation of retrieval-augmented generation strategies, revealing a critical robustness gap in current LLMs.
• We identify a significant trade-off between domain adaptation and safety. Our experiments with continued pretraining show that while domain-specific training improves standard accuracy, it degrades the models' uncertainty awareness.

Table 1: The statistics of the SEARCHFIRESAFETY dataset.

| Category | Statistic | Number |
|----------|-----------|--------|
| Open-Ended QA | Total pairs | 941 |
| | Pairs with mapped documents | 702 |
| | Avg. question (answer) length | 97.14 (267.39) |
| | Avg. relevant documents per question (excluding zeros) | 1.13 (1.52) |
| Legal Documents | Total documents | 4437 |
| | Avg. length in each document | 478.84 |
| | Avg. words in each document | 103.37 |
| | Avg. relevant documents (excluding zeros) | 1.84 (4.71) |
| Single-Hop QA (Yes/No) | Total pairs | 9238 |
| Multi-Hop QA (MCQ) | Total pairs | 4007 |

## 2 REAL-WORLD OPEN-ENDED QA

The primary goal of this work is to construct a question answering (QA) dataset grounded in real-world scenarios derived from fire safety legislation and requiring legal reasoning.

**Data Collection** We crawled the official government petition portal of the Korean National Fire Agency (NFA) to gather QA records published between February 23, 2023, and April 30, 2025.[1] Each record contains a citizen inquiry and the corresponding official response from an NFA officer; we treat the official response as the **gold-standard** answer. From these records, we parsed 941 single-row QA instances.

NFA officers cite relevant legal documents explicitly in their responses. To link each question to its supporting legal documents, we first employed BM25 (Robertson & Zaragoza, 2009) to generate candidate pairings between the legal sources referenced in NFA answers and the titles in our compiled legal corpus. Subsequently, all authors independently reviewed each QA instance alongside its candidate statutes in a side-by-side viewer to verify and finalize these mappings.

**Statistics** Table 1 (upper block) summarizes the dataset: it contains 941 Korean QA pairs with average question and answer lengths of 97.14 and 267.39 characters, respectively. Among these, 702 questions are linked to at least one supporting document, yielding an average of 1.13 linked documents per question (1.52 when excluding unmapped cases).

**Data Analysis** An illustrative QA pair appears in the top block of Table 2. The real-world, open-ended QA subset has two properties that make retrieval challenging. First, because the questions are posed by non-experts, there is a persistent gap between colloquial phrasing and formal legal terminology. This linguistic mismatch complicates retrieval—especially sparse methods—since everyday expressions (e.g., *outdoor fire equipment*) may refer to narrowly defined statutory terms (e.g., *outdoor hydrant*), undermining exact lexical matching and even semantic linkage.

Second, the questions are distributed across four broad categories (see Appendix B). Notably, 15.7% of questions fall into the Interpretation of Regulations category. These are queries that directly reference legal statutes by name or number, such as, "Does Article 19 of the Building Act not apply?" The prevalence of these explicit citations presents an opportunity to improve retrieval. To capitalize on this, we prepend structured metadata—specifically the law name and article identifier—to each legal document, allowing retrievers to better match these precise references.

---

[1] https://www.epeople.go.kr/

Table 2: An example from SEARCHFIRESAFETY. Comprising a real-world inquiry and the official response issued by the Korean National Fire Agency (NFA). The answer is grounded in a specific legal provision, linked via the corresponding `Matched Document ID` (red). Each matched document may also reference `Related Document IDs` (blue), indicating cross-referenced provisions.

---

**Question ID:** 49
**Question:** I would like to inquire whether a removable safety railing installed in a school, with an installation height exceeding 1.2 meters, can still be recognized as an opening.
**Answer:** According to Article 2, Subparagraph 1, Item (b) of the Enforcement Decree of the Act on the Installation and Management of Fire-Fighting Systems, the height of an opening is defined as the vertical distance from the floor to the bottom of the opening, and it shall be no more than 1.2 meters. Therefore, if the height of a removable safety railing exceeds 1.2 meters, the area shall be regarded as a windowless floor.

---

**Matched Document ID:** 3057
**Matched Document:** Article 2 (Definitions) The terms used in this Decree are defined as follows:
1. A "windowless floor" means a ground floor with an opening meeting all the following conditions (referring to window and entrance, created for lighting, ventilation, air circulation, entrance, etc., other similar things; hereinafter the same shall apply) whose aggregate floor area does not exceed 1/30 of the total area (referring to the area calculated pursuant to Article 119 (1) 3 of the Enforcement Decree of the Building Act; hereinafter the same shall apply):
   a. It shall be big enough for a circle with at least 50 centimeters in diameter can pass through;
   b. It shall be at least 1.2 meters high from the surface of the floor to the bottom of its opening; (...)

---

**Related Document ID:** 2027
**Related Document:** Article 119 (Methods of Calculating Area)
(1) Pursuant to Article 84 of the Act, the area, height, and number of floors of a building shall be calculated as follows: (...)
   3. Floor area means the area of the horizontal projection plane of each floor of a building or part of the building enclosed by the centerlines of walls, columns, or other similar partitions; (...)

---

## 3 LEGAL DOCUMENT CONSTRUCTION

We aim to build and release a temporally current corpus of Korean statutes and subordinate regulations. Because these legal documents evolve through frequent amendments, an automated pipeline capable of continuous updates is essential. To this end, we implemented a crawler for the Korea National Law Information Center[2] to construct a corpus reflecting all laws and regulations in force as of April 30, 2025.

**Citation Graph Construction via Hyperlinks**  As illustrated in Table 2, a single parent instrument rarely contains all relevant information; instead, it delegates to subordinate statutes, enforcement rules, or notices via links to *related documents*. To capture inter-document dependencies that support multi-hop retrieval, we parse `<a>` tags in statutory HTML pages and treat each intra-corpus hyperlink to another statute or regulation as a directed edge. Post-processing removes malformed or external links and normalizes anchors to canonical provision identifiers.

**Human-in-the-loop Curation**  While assembling the corpus, we encountered two significant issues that impede machine readability. First, detailed provisions like annexes are often provided as standalone PDF files. Second, essential artifacts within the primary HTML, such as tables and mathematical formulas, are frequently embedded as images rather than machine-readable text. To address this, we built an automated ingestion pipeline augmented with a human-in-the-loop (HITL) verification stage:

---

[2] https://www.law.go.kr/

- PDF Extraction: For supplementary PDF documents, we manually downloaded the files and then used GPT-4o together with `pdfplumber`[3] to extract text.
- Image Transcription: For the 2% of provisions containing content embedded as images, we employed GPT-4.1-mini to transcribe visual elements into structured text.

Both outputs were subsequently audited by human annotators, who corrected transcription errors and ensured fidelity to the source material.

**Chunking Strategy and Statistics**   To preserve legal semantics during indexing, we align chunks with native legal units. For *statutes*, we chunk at the Article level; for *administrative rules*, we use second-level decimal headings (e.g., 1.1); and for *annex tables* ("byeolpyo"), we chunk at the item ("ho") level. This unit-aware segmentation minimizes cross-provision fragmentation while supporting fine-grained retrieval. We also prepend metadata—specifically the law name and article identifier—to each chunked document.

The final corpus comprises 4,437 legal documents—spanning statutes, enforcement decrees/rules, and administrative notices. Documents contain, on average, 478.84 Korean characters (approximately 103.37 words). Each document links to an average of 1.84 other documents; excluding isolates, the average rises to 4.71, indicating substantial inter-document connectivity.

**Corpus Coverage**   Our corpus construction is guided by two key design principles: *practical relevance* and *structural connectivity*. First, by curating sources cited frequently in NFA responses, we concentrate on "active legislation"—provisions that actually trigger inquiries in real-world scenarios. This approach avoids diluting the benchmark with dormant or rarely applied laws, ensuring the corpus reflects the *working knowledge* required for genuine legal consultation. Second, rather than treating these statutes as isolated texts, we explicitly trace and preserve their citation links to form a cohesive legal graph. Unlike approaches that stochastically sample unrelated laws, our method retains the valid legislative dependencies essential for interpretation. This combination of high-utility content and preserved structure allows us to model realistic legal contexts, serving as the necessary foundation for the citation-graph-based synthetic data generation described in the subsequent section.

## 4   SYNTHETIC QA CONSTRUCTION AND HALLUCINATION

While the open-ended tasks described in the previous section provide realistic data points, they present significant challenges for consistent quantitative evaluation. To address this limitation and rigorously assess legal reasoning capabilities, we construct a synthetic evaluation set designed to explicitly probe model hallucination. Hallucination remains a critical barrier to practical deployment in legal domains where trustworthiness is paramount (Magesh et al., 2024). To quantify this failure mode, we formulate our benchmark as a multiple-choice question (MCQ) task, encompassing both standard queries and yes/no questions.

We constructed this dataset as a multi-hop question answering task rooted in the citation graph topology introduced in Section 3. Instead of sampling arbitrary documents, we generated questions from pairs of documents (Document A and Document B) explicitly linked within the graph. The fundamental design principle is strict conditional dependency: questions are constructed to be unanswerable from the primary document (Document A) in isolation, becoming solvable only when synthesized with the referenced document (Document B). An illustrative example of such an MCQ is presented in Table 3.

Accordingly, each item consists of a naturally phrased question without explicit citation markers, accompanied by five options: one correct answer derivable only from the combination of Documents A and B, one uncertainty option (e.g., "Cannot be determined"), and three plausible distractors. This structure allows us to effectively detect whether a model hallucinates an answer based on insufficient context.

---

[3]https://pypi.org/project/pdfplumber/

Table 3: Example synthetic multiple-choice QA constructed from Document 3057 and Document 2027 in Table 2, illustrating that the correct choice is identifiable only under full context.

---

**Question:** When checking whether the total opening area stays within 1/30 of the floor area, which definition of *floor area* should be used?

**Option 1:** The gross area measured by the outermost exterior dimensions of the building.

**Option 2:** The horizontal projected area of each floor enclosed by the *centerlines of walls, columns, or similar partitions*.

**Option 3:** The usable interior area excluding all walls, columns, and service shafts.

**Option 4:** The sum of areas of all rooms shown on the interior finish plan.

**Option 5:** Cannot be answered with the given information.

**Correct Answer (Full Context):** Option 2

**Correct Answer (Partial Context):** Option 5

**Rationale:** The area-calculation rule needed to interpret "floor area" is present only in the related document. With full context, the correct definition is the centerline-based horizontal projection (Option 2). With partial context, the definition is missing, so the question is not answerable (Option 5).

---

Using GPT-4o (OpenAI, 2024), we synthetically generated an initial set of 5,091 MCQ pairs. To ensure data quality, human annotators conducted an exhaustive review, resulting in a final validated set of 4,007 questions. Items were discarded based on three primary criteria: a small subset (3 items) contained malformed answer sets; a modest number (55 items) remained unanswerable even with both documents; and the majority of excluded items (1,076 items) failed the dependency criterion, as they were answerable using Document A alone despite being designed for multi-hop reasoning.

## 5 EXPERIMENTS

### 5.1 EXPERIMENTAL SETUP

**Models** We evaluate five publicly available LLMs with Korean capability. These include Qwen3-8B (Team, 2025); Exaone3.5-2.4B and 7.8B (LG AI Research, 2024); HyperClova-1.5B; and GPT-4o (OpenAI, 2024). All open-weight models are run in FP16 on a single RTX-A6000 (48GB), whereas GPT-4o is accessed through the OpenAI API. For Qwen3-8B, we utilized a reasoning mode.

**Evaluation Protocols** We evaluate the Multi-Hop QA dataset under three complementary settings, each isolating a different capability of the RAG pipeline:

1. **Zero-Shot (no context).** The model is given only the question, without any supporting documents. This setting measures parametric knowledge.
2. **Full Context (gold context; Doc A+B).** The model is provided with the full gold context. For instance, in the MCQ task, this encompasses both Document A and Document B. Since each instance is designed such that the answer can only be derived by synthesizing information from both documents, this setting evaluates multi-hop reasoning under ideal evidence conditions.
3. **Partial Context (Doc A only).** The model receives Document A together with the question, while Document B is withheld. The prompt explicitly includes an additional option, *"Cannot be determined with the given information"*, and instructs the model to select it when evidence is insufficient. Because Partial Context examples are unanswerable by design, this setting evaluates both (i) reasoning over incomplete context and (ii) uncertainty awareness—i.e., the ability to abstain instead of hallucinating (see Table 7).

Table 4: Generation performance (%) on real-world open-ended QA across four retrieval strategies: Zero-Shot (no context) and Full Context (gold context). **Bold** = best within each model.

| Model | Strategy | ROUGE-1 | ROUGE-L | BERTScore | LLM-AS-A-JUDGE | WIN-RATE |
|---|---|---|---|---|---|---|
| HyperCLOVA-1.5B | Zero-Shot | 22.57 | 20.13 | 62.85 | 6.84 | 6.13 |
| | Full Context | **28.47** | **25.68** | **64.84** | **26.64** | **8.97** |
| Exaone3.5-2.4B | Zero-Shot | 31.09 | 27.13 | 55.26 | 6.55 | 9.54 |
| | Full Context | **41.87** | **37.09** | **60.08** | **13.96** | **11.97** |
| Exaone3.5-7.8B | Zero-Shot | 28.64 | 24.68 | 55.59 | 13.96 | **15.10** |
| | Full Context | **42.84** | **38.50** | **61.62** | **47.29** | 13.53 |
| Qwen3-8B | Zero-Shot | 27.24 | 23.29 | 55.86 | 11.11 | 11.54 |
| | Full Context | **43.49** | **38.96** | **59.70** | **17.95** | **17.38** |
| GPT-4o | Zero-Shot | 20.91 | 18.61 | 59.74 | 24.50 | 15.95 |
| | Full Context | **28.60** | **26.49** | **66.30** | **58.97** | **17.52** |

**Evaluation Metrics**  To evaluate open-ended generation, we report a combination of reference-based and model-based metrics. First, we compute lexical- and embedding-level overlap with the gold answer using ROUGE-1/L (Lin, 2004) and BERTScore (Zhang et al., 2020). While informative, these metrics may not fully capture semantic correctness, especially when multiple valid phrasings exist. For multiple-choice tasks (Multi-hop MCQA and Single-hop Yes/No QA), we report top-1 ACCURACY (%).

To assess factual and semantic alignment more directly, we adopt an LLM-AS-A-JUDGE protocol (Liu et al., 2023). Specifically, we use GPT-4o OpenAI (2024) as the evaluator. For each instance, the judge is provided with the question, the gold answer, and the model's response, and is instructed to output a binary decision indicating whether the response is correct with respect to the gold answer. The exact evaluation prompt and decision rubric are fixed across all experiments and are provided in Appendix I. In addition, we report WIN-RATE in a pairwise comparison setting (Wang et al., 2024; Wolfe, 2023), where GPT-4o selects the better of two answers: the model output versus the gold answer. Win-Rate is defined as the proportion of cases in which the model output is preferred by the judge.

### 5.2 REAL-WORLD OPEN-ENDED QA RESULTS

**Generation Performance**  Table 4 reports generation results on the real-world open-ended QA. Because ROUGE-1/L and BERTScore primarily reflect lexical/semantic similarity to human references rather than factual correctness, we treat them as auxiliary indicators. Even so, conditioning on legal context consistently increases similarity to human answers compared to answering directly. For example, for Exaone3.5-7.8B, moving from *Zero-Shot* to *Full Context* raises ROUGE-L from 27.13 to 38.50 and BERTScore from 55.59 to 61.62.

Despite achieving high lexical similarity to human references, models like Exaone3.5-2.4B and Qwen3-8B receive low LLM-AS-A-JUDGE scores (13.96 and 17.95, respectively). This discrepancy indicates that their outputs, while superficially plausible, often contain conclusions that diverge markedly from expert legal interpretations. Even with *Full Context* documents, state-of-the-art GPT-4o falls short of domain-expert gold answers by either metric: LLM-AS-A-JUDGE (58.97%) and WIN-RATE (17.52%).

**LLM Judge Reliability**  In Table 5, we further assess the reliability of LLM-AS-A-JUDGE by conducting a human evaluation of GPT-4o's answers. Overall agreement between the LLM judge and human raters is high at 88.30% (TP+TN). Nevertheless, false negatives account for 10.80% of cases, which suggests that the LLM judge is more stringent than human annotators. For example, it may label an answer as *incorrect* when it is

Table 5: Confusion Matrix between LLM-AS-A-JUDGE predictions and HUMAN ANNOTATION. TP=61.20%, FP=0.90%, FN=10.80%, TN=27.10%.

| | HUMAN ANNOTATION | |
| --- | --- | --- |
| LLM-AS-A-JUDGE | Correct | Incorrect |
| Correct | 61.20 | 0.90 |
| Incorrect | 10.80 | 27.10 |

factually consistent yet underspecified, whereas a human rater would deem it *correct* (refer to Appendix K for further analysis).[4]

**Case Study: Hallucination in Legal Reasoning**   The most revealing insight comes from the true negatives (i.e., TN=27.10%). A substantial portion of these cases highlights the inherent difficulty of the legal reasoning task itself, a challenge that persists even with full access to relevant documents. A common failure mode is the model's inability to connect colloquial user phrasing with precise statutory terminology, leading it to invert conclusions about legal responsibility.

For example, consider the query: "Must a *tamper switch* be installed on the shutoff valve of the indoor fire-hydrant water-supply pipe?" The applicable regulation states that "the shutoff valve · · · must provide an open/close indication." Models often fail to recognize that a *tamper switch* is the specific device that provides this indication. This leads them to the incorrect conclusion that "Although the regulation requires an open/close indication, there is no explicit rule requiring a *tamper switch*; therefore, installation is not mandatory."

## 5.3 SYNTHETIC MULTIPLE CHOICE QA RESULTS

**Parametric vs. External Knowledge**   Table 6 compares performance between the *Zero-Shot* and *Full Context* setting. The Zero-Shot setting relies solely on a model's internal (parametric) knowledge, whereas the Full Context setting provides all required information—Document A plus Document B—to answer each question. With complete information, most models achieve substantial gains over their Zero-Shot baselines. Notably, Exaone3.5-7.8B (77.31%) and Qwen3-8B (74.91%) slightly outperform GPT-4o (73.26%) under Full Context. This pattern suggests that, while GPT-4o is strong at recognizing when information is missing, other models can be highly effective at synthesizing evidence when the relevant context is fully provided.

To understand how context changes model behavior, we analyze instances where a model's prediction flips between settings. The *Correction Rate*—the proportion of Zero-Shot errors corrected under Full Context—highlights the benefit of retrieval: most models correctly revise between 53.87% and 64.66% of their initial mistakes. However, even accurate context can sometimes harm performance. We observe *context-induced errors*, quantified as the *Introduced Error Rate* (IER), where a previously correct Zero-Shot answer becomes incorrect after conditioning on the provided documents. IER is highest for GPT-4o at 18.68% and is also notable for Exaone3.5-7.8B at 12.05%. These effects align with prior observations that LLMs can struggle to blend contextual knowledge with parametric knowledge (Xu et al., 2024).

**Uncertainty Awareness**   The *Partial Context* setting requires models to recognize that the provided documents are insufficient to determine the answer and to abstain accordingly. However, LLMs often lack

---

[4]To validate this comparison, two authors independently rated the model outputs. Because each question was paired with the NFA's official answer (i.e., the gold standard) and the relevant legal documents were provided explicitly, reliable evaluation was feasible even without specialized legal expertise. Accordingly, we obtained a high Cohen's Kappa score ($\kappa = 0.88$), indicating strong inter-rater agreement. Any remaining disagreements were resolved through discussion until a consensus was reached.

Table 6: Accuracy (%) on the Multi-Hop QA dataset for Zero-Shot and Full Context scenarios. **Zero-Shot** evaluates parametric knowledge, while **Full Context** (Doc A+B) evaluates reasoning with complete information. We also report accuracy changes conditioned on the initial Zero-Shot prediction.

| Model | Zero-Shot | Full Context | Correction Rate | Introduced Error Rate |
|---|---|---|---|---|
| HyperCLOVA-1.5B | 53.19 | 69.16 | 53.87 | 17.39 |
| Exaone3.5-2.4B | 55.34 | 74.43 | 58.59 | 12.78 |
| Exaone3.5-7.8B | 55.39 | **77.31** | 64.09 | **12.05** |
| Qwen3-1.7B | 31.54 | 45.56 | 36.98 | 35.82 |
| Qwen3-8B | 53.96 | 74.91 | **64.66** | 16.35 |
| GPT-4o | **59.94** | 73.26 | 61.20 | 18.68 |

Table 7: Accuracy (%) on the Partial Context in Multi-Hop QA.

| Model | Partial Context |
|---|---|
| HyperCLOVA-1.5B | 8.82 |
| Exaone3.5-2.4B | 45.86 |
| Exaone3.5-7.8B | 53.69 |
| Qwen3-1.7B | 71.98 |
| Qwen3-8B | 51.66 |
| GPT-4o | **72.73** |

Table 8: Accuracy (%) on the Single-hop QA dataset.

| Model | Zero-Shot | Full Context |
|---|---|---|
| HyperCLOVA-1.5B | 39.67 | 91.65 |
| Exaone3.5-2.4B | 76.93 | 94.17 |
| Exaone3.5-7.8B | 77.02 | **96.50** |
| Qwen3-8B | 53.03 | 90.15 |
| GPT-4o | **79.60** | 96.29 |

calibrated awareness of what they do not know and tend to answer indiscriminately (Zhao et al., 2024). In our experiments, GPT-4o exhibits the strongest uncertainty awareness, achieving 72.73% accuracy in detecting information insufficiency (Table 7). By contrast, smaller models—most notably HyperClova-1.5B at 8.82%—struggle and frequently attempt to answer despite incomplete evidence. Qwen3-8B (51.66%) and Exaone variants (45.86%, 53.69%) also perform poorly. These results underscore how difficult it is, in the legal domain, for models to identify noisy or incomplete support and to refrain from overconfident generation.

**Single-Hop QA Results**   Table 8 demonstrates that providing the relevant legal document significantly improves accuracy across all models. GPT-4o achieves the highest Zero-Shot performance (79.60%), while Exaone-7.8b reaches the top accuracy with Oracle RAG (96.50%). The most substantial improvement is observed in HyperClova-1.5B, which jumps from 39.67% to 91.65%. This highlights that while the model may lack extensive internalized legal knowledge, it possesses strong reading comprehension capabilities when grounded in the correct statute.

**Training on a Legal-Domain Corpus**   To study domain adaptation effects, we perform continued pretraining (CPT) for Qwen3-8B using a legal-domain corpus. The training data combines our collected fire-safety statutes with additional Korean legal resources from AI Hub[5] (criminal law, civil law, and intellectual property law), totaling 0.83B tokens.

Table 9 compares the base model to its CPT-adapted counterpart. CPT improves standard accuracy, increasing Zero-Shot by +4.93%p and Full Context by +3.77%p, bringing performance close to GPT-4o under Full Context. However, CPT substantially reduces Partial Context accuracy from 51.66% to 38.94% (-12.72%p). Since Partial Context questions are unanswerable by design, this drop indicates weaker uncertainty awareness and a higher tendency to produce answers under insufficient evidence. These results highlight a non-trivial

---

[5] https://www.aihub.or.kr/

Table 9: Comparison between Qwen3-8B and Qwen3-8B + CPT (continued pre-training) on Multi-Hop QA under Zero-Shot, Partial Context, and Full Context settings.

| Model | Zero-Shot | Partial Context | Full Context |
|---|---|---|---|
| Qwen3-8B | 53.96 | 51.66 | 74.91 |
| Qwen3-8B + CPT | 58.89 (+4.93) | 38.94 (-12.72) | 78.68 (+3.77) |

trade-off between accuracy under complete evidence and robustness to missing evidence, reinforcing the value of SEARCHFIRESAFETY in revealing such failure modes beyond conventional single-setting QA evaluation.

### 5.4 SUMMARY OF FINDINGS

Across both real-world open-ended QA and synthetic multi-hop MCQA, our results highlight three consistent patterns. First, providing relevant legal documents substantially improves answer quality, confirming that external grounding is indispensable for legal QA. Second, performance under Full Context can be strong even for mid-sized models, but this strength does not transfer to Partial Context settings, where many models fail to abstain and instead hallucinate. Third, domain-adaptive training via continued pretraining improves standard accuracy yet can reduce uncertainty awareness, indicating that progress measured by conventional QA metrics may conceal important safety regressions. Together, these findings validate the central motivation of SEARCHFIRESAFETY: legal RAG systems must be evaluated not only for correctness under complete evidence but also for robustness and calibrated abstention under missing or noisy evidence.

## 6 RELATED WORK

The NLP community has shown growing interest in the legal domain (Ariai & Demartini, 2024). Previous studies, such as LexGLUE (Chalkidis et al., 2022; Niklaus et al., 2023), have demonstrated the applicability of language models to a range of legal tasks, including judgment prediction and question answering. With the rapid advancement of LLMs, legal retrieval datasets have also emerged across multiple jurisdictions and languages (Louis & Spanakis, 2022; Zhong et al., 2020; Liu et al., 2024; Hou et al., 2025; Pipitone & Alami, 2024; Gao et al., 2024). For instance, CLERC (Hou et al., 2025) compiles U.S. federal case documents and links citation data to support reference retrieval and long-form answer generation. Recent efforts such as Zheng et al. (2025) further demonstrate the growing interest in developing high-quality legal RAG datasets. Non-English datasets include the French statutory retrieval benchmark BSARD (Louis & Spanakis, 2022) and Chinese legal retrieval datasets such as LeDQA (Liu et al., 2024) and JEC-QA (Zhong et al., 2020). In the Korean legal domain, LEGAR-BENCH (Kim et al., 2025) focuses on legal case retrieval, while LBOX-Open (Hwang et al., 2022) provides multi-task annotations—such as classification, judgment prediction, and summarization—within legal case documents.

## 7 CONCLUSION

We introduce SEARCHFIRESAFETY, the first Korean QA dataset for retrieval-augmented generation in fire-safety law, combining real-world open-domain queries, synthetic single-hop and multi-hop tasks, authoritative legal documents, and an explicit citation graph to evaluate retrieval, generation, reasoning, and *uncertainty awareness*. Experiments indicate that stronger retrieval substantially improves factual grounding when relevant context is supplied, yet multi-step legal reasoning remains challenging. We expect this work to catalyze research in legal AI by providing a realistic, regulation-heavy benchmark for this challenging domain.

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

## A    DISCUSSION

**Usefulness of RAG in the Legal Domain**    Fine-tuning (FT) versus Retrieval-Augmented Generation (RAG) remains an active debate in NLP. Recent studies suggest that, for knowledge injection, RAG often outperforms FT for models under 10B parameters (Soudani et al., 2024; Ovadia et al., 2024). Moreover, in the legal profession, the case for RAG is even stronger: statutes and regulations evolve continuously, and provisions frequently cross-reference or delegate to subordinate instruments. Maintaining *temporal currency* and *citation awareness* therefore requires retrieval over up-to-date sources rather than static parametric memories. This motivates the kind of continuously maintainable data pipeline we propose. On the other hand, our results reveal limitations of RAG: even state-of-the-art models struggle when tasks demand synthesizing information across multiple, subtly conflicting provisions—a hallmark of genuine legal analysis. Thus, RAG is necessary but insufficient; legal QA also needs explicit mechanisms for conflict resolution, terminology grounding, and calibrated abstention.

**Retrieval Performance Is Key**    Improving LLM performance in law is primarily a retrieval problem. Real-world, open-ended questions exhibit a large semantic gap between lay phrasing and formal legal terminology, which makes recall difficult and precision brittle. Consistent with this pattern, when supplied with *gold* context, Exaone3.5-7.8B and Qwen3-8B outperform GPT-4o on accuracy; even a 2.4B model surpasses GPT-4o in some settings—reinforcing evidence that small models can be competitive agents when context is reliable (Belcak et al., 2025). Yet in our uncertainty-awareness evaluation, smaller models are far more prone to answer unconditionally, even when context is incomplete or noisy. This aligns with findings that feeding incorrect documents does not reliably increase—and can even *decrease*—uncertainty (Soudani et al., 2025). LLMs tend to *lock in* to provided context, and, as Kalai et al. (2025) note, binary grading regimes can still reward guessing when retrieval fails to yield a confident answer.

## B    REAL-WORLD OPEN-ENDED QA

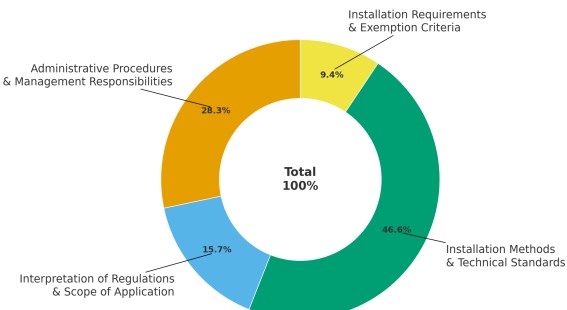

Figure 2: Distribution of Inquiry Types on Real-World Open-Ended QA.

Figure 2 summarizes the distribution of inquiry types across four regulatory domains. Nearly half of all queries (46.6%) concern installation methods and technical standards, indicating that practitioners most frequently seek granular, practice-oriented guidance to resolve on-site implementation issues. Administrative procedures and management responsibilities account for a further 28.3%, reflecting sustained demand for clarity on permitting, documentation, inspection protocols, and accountability frameworks. A smaller, yet substantive, proportion (15.7%) pertains to the interpretation of regulations and the scope of application, including the hierarchical resolution of conflicting criteria and the explication of defined legal terms. Finally, inquiries about installation requirements and exemption criteria comprise 9.4%, typically probing the conditions under which fire-protection measures are mandatory, substitutable, or waivable.

## C   RETRIEVAL PERFORMANCE

Table 10: Retrieval performance of different strategies and methods on real-world open-ended QA.

| Strategy | Method | Language | Recall@1 | Recall@5 | Recall@10 | Recall@100 | MRR |
|---|---|---|---|---|---|---|---|
| Sparse | TF-IDF | - | 7.85 | 17.81 | 23.18 | 51.40 | 15.47 |
|  | BM25 | - | 8.90 | 17.39 | 22.65 | 48.33 | 16.14 |
| Dense | MiniLM-L6 | English | 0.05 | 0.19 | 0.33 | 3.17 | 0.37 |
|  | KR-SBERT | Korean | 4.67 | 10.70 | 16.12 | 46.90 | 10.92 |
|  | Qwen3-emb | Multilingual | 15.72 | 38.00 | 48.23 | 77.83 | 31.55 |
|  | BGE-m3 | Multilingual | **19.54** | 42.52 | 53.00 | 80.65 | 35.94 |
| HyDE | BGE-m3 | Multilingual | 13.96 | 38.15 | 47.44 | 78.07 | 30.74 |
| Hybrid | RRF | - | 13.03 | 37.57 | 49.67 | 79.70 | 29.40 |
|  | wRRF | - | **19.54** | **42.71** | **53.19** | **80.98** | **36.12** |

**Evaluation Metrics**   Retrieval effectiveness is measured using two standard metrics: Recall@K, which calculates the proportion of queries for which relevant documents are retrieved within the top-K results, and Mean Reciprocal Rank (MRR), which emphasizes early accuracy by averaging the reciprocal ranks of the first relevant document.

**Results**   Table 10 reports results on our real-world, open-ended Korean QA dataset. Overall, *dense retrievers* substantially outperform sparse methods such as TF–IDF and BM25. In real-world settings, non-expert users often use vocabulary that differs markedly from the terminology used in legal documents; as a result, many queries provide few lexical cues for sparse retrieval.

Our dataset contains many documents with numerals, Sino-Korean expressions, and mixed scripts, which tends to favor multilingual encoders over monolingual ones. Consistent with this, multilingual models (e.g., BGE-m3, Qwen3-emb) surpass MiniLM-L6 (English-only) and KR-SBERT (Korean-only). The HyDE strategy did not improve over using BGE-m3 alone (e.g., Recall@1=13.96 and MRR=30.74). Naive hybrid approaches that combine sparse and dense signals degraded performance—likely due to the weak sparse component—reducing Recall@1 to 13.03% and MRR to 29.40%. By contrast, our proposed hybrid strategy, weighted RRF, achieved small but consistent gains: Recall@100 improved from 80.65% (BGE-m3 alone) to 80.98%, and MRR increased from 35.94% to 36.12%.

## D   WEIGHTED RECIPROCAL RANK FUSION

**Retrieval Strategy**   For *sparse* retrieval, we use TF–IDF (Salton & Buckley, 1988) and BM25 (Robertson & Zaragoza, 2009), indexing 3-grams over Hangul Jamo–decomposed text. For *dense* retrieval, we evaluate MiniLM-L6[6], KR-SBERT (Park & Shin, 2021), Qwen3-emb (Zhang et al., 2025), and BGE-m3 (Chen et al., 2024a).

We also assess advanced strategies, including Hypothetical Document Embeddings (HyDE) (Gao et al., 2023), which generates a synthetic document from the query and uses its embedding for retrieval. Additionally, we evaluate *hybrid* methods that combine sparse and dense results via Reciprocal Rank Fusion (RRF) (Cormack et al., 2009). Alongside the standard formulation, we test a weighted RRF (1:9 sparse-to-dense ratio) to better reflect observed real-world query distributions (see Appendix D).

---

[6]https://huggingface.co/sentence-transformers/all-MiniLM-L6-v2

**Weighted Reciprocal Rank Fusion**    To integrate results from both sparse and dense retrievers, we adopt *Reciprocal Rank Fusion (RRF)* (Cormack et al., 2009), a simple yet effective method for combining ranked lists from multiple retrieval models. RRF is attractive because it avoids dependence on the raw similarity scores of individual systems, which are often not directly comparable across models. Instead, it relies only on rank positions, making it robust across heterogeneous retrieval methods. Formally, given a query $q$, a candidate document $d$, and a set of retrieval models $M$, the RRF score is defined as:

$$RRF(q, d, M) = \sum_{m \in M} \frac{1}{k + \pi^m(q, d)}, \tag{1}$$

where $\pi^m(q, d)$ denotes the rank of $d$ under model $m$. The constant $k$ is a smoothing parameter that reduces the dominance of very highly ranked documents from any single model. By construction, RRF ensures that a document ranked moderately well by multiple systems can receive a higher fused score than a document ranked extremely high by only one system.

While RRF offers a simple and robust mechanism for combining heterogeneous retrieval models, it treats all models equally regardless of their effectiveness for a given task. This uniform treatment can be suboptimal in domains where the relative utility of sparse and dense retrievers varies significantly across query types. To address this limitation, we propose *Weighted Reciprocal Rank Fusion (wRRF)*, a novel extension of RRF that assigns an explicit weight $w_m$ to each model $m \in M$:

$$wRRF(q, d, M) = \sum_{m \in M} w_m \cdot \frac{1}{k + \pi^m(q, d)}, \quad \text{subject to} \sum_{m \in M} w_m = 1, \ w_m \geq 0. \tag{2}$$

By explicitly controlling the contribution of each model, WRRF enables a more flexible and task-aware integration of sparse and dense retrievers, while preserving the robustness of the original RRF formulation.

**Hyperparameter Choices**    WRRF introduces two key hyperparameters: the smoothing constant $k$ and the model weights $w_m$. Unlike prior work, which commonly fixes $k = 60$, we empirically found that a smaller constant provides more stable performance in our domain-specific evaluation. Large values of $k$ down-weight top ranks too heavily, leading to less discriminative results. We therefore set $k = 5$ for all experiments, which emphasizes the contribution of top-ranked items while still maintaining balance across models. This choice was especially effective in the legal domain, where queries often correspond to highly specific information needs and relevant documents are typically concentrated at the top of each retriever's ranking.

For the model weights, we relied on the query type distribution analyzed in Appendix B. Our analysis shows that roughly 15% of queries explicitly mention statutes or legal provisions, while the remaining majority require semantic reasoning over legal texts without explicit references. Based on this distribution, we adopted a 1:9 weighting scheme between sparse and dense retrievers, assigning $w_{\text{sparse}} = 0.1$ and $w_{\text{dense}} = 0.9$. This configuration reflects the empirical query composition, preserving the strength of sparse retrieval for explicit law mentions while relying primarily on dense retrieval for the majority of queries. By combining a smaller $k$ with task-informed weighting, WRRF captures the complementary strengths of sparse and dense retrievers and improves robustness in real-world open-ended QA scenarios.

## E  FULL RETRIEVAL PERFORMANCE RESULTS

Table 11: Results on Real-World Open-Ended QA

| Method | Recall@1 | Recall@3 | Recall@5 | Recall@10 | Recall@20 | Recall@50 | Recall@100 | nDCG@1 | nDCG@3 | nDCG@5 | nDCG@10 | nDCG@20 | nDCG@50 | nDCG@100 |
|---|---|---|---|---|---|---|---|---|---|---|---|---|---|---|
| TF-IDF | 7.85 | 12.88 | 17.81 | 23.18 | 29.98 | 43.36 | 51.40 | 9.53 | 11.35 | 13.44 | 15.29 | 17.10 | 19.95 | 21.34 |
| BM-25 | 8.90 | 13.76 | 17.39 | 22.65 | 29.11 | 40.04 | 48.33 | 10.81 | 12.37 | 13.95 | 15.70 | 17.46 | 19.78 | 21.23 |
| MiniLM-L6 | 0.05 | 0.19 | 0.19 | 0.33 | 0.55 | 1.75 | 3.17 | 0.14 | 0.17 | 0.17 | 0.21 | 0.27 | 0.53 | 0.77 |
| KR-SBERT | 4.67 | 8.47 | 10.70 | 16.12 | 23.40 | 34.72 | 46.90 | 6.12 | 7.53 | 8.45 | 10.24 | 12.11 | 14.47 | 16.57 |
| Qwen3-emb | 15.72 | 29.16 | 38.00 | 48.23 | 57.35 | 68.65 | 77.83 | 19.77 | 25.26 | 28.95 | 32.45 | 34.92 | 37.33 | 38.98 |
| BGE-m3 | **19.54** | _33.95_ | _42.52_ | _53.00_ | **62.53** | _73.07_ | _80.65_ | **23.76** | _29.60_ | _33.27_ | _36.81_ | _39.42_ | _41.71_ | _43.07_ |
| RRF | 13.03 | 29.34 | 37.57 | 49.67 | 59.30 | 71.86 | 79.70 | 16.07 | 23.69 | 27.21 | 31.28 | 33.91 | 36.68 | 38.08 |
| wRRF | **19.54** | **34.38** | **42.71** | **53.19** | **62.53** | **74.11** | **80.98** | **23.76** | **29.83** | **33.38** | **36.97** | **39.54** | **42.05** | **43.30** |

Table 12: Results on Synthetic Multi-Hop QA

| Method | Recall@1 | Recall@3 | Recall@5 | Recall@10 | Recall@20 | Recall@50 | Recall@100 | nDCG@1 | nDCG@3 | nDCG@5 | nDCG@10 | nDCG@20 | nDCG@50 | nDCG@100 |
|---|---|---|---|---|---|---|---|---|---|---|---|---|---|---|
| TF-IDF | 20.20 | 37.00 | 44.57 | 52.68 | 59.76 | 67.50 | 72.53 | 40.25 | 36.74 | 40.54 | 43.78 | 45.98 | 47.88 | 48.88 |
| BM-25 | 26.20 | 41.79 | 47.49 | 53.78 | 59.70 | 65.69 | 69.93 | 52.16 | 43.35 | 46.24 | 48.75 | 50.59 | 52.05 | 52.89 |
| Qwen3-emb | 34.88 | _57.98_ | 65.48 | 73.22 | _79.56_ | _86.08_ | _89.61_ | 69.54 | 59.47 | 63.27 | 66.35 | 68.32 | 69.92 | 70.63 |
| BGE-m3 | **35.82** | 57.73 | 65.24 | 72.85 | 79.00 | 85.52 | 89.40 | **71.34** | _59.73_ | _63.53_ | _66.57_ | _68.49_ | _70.10_ | _70.87_ |
| RRF | 34.56 | 57.45 | **65.94** | **73.25** | 79.24 | 85.36 | 88.96 | 68.79 | 58.87 | 63.16 | 66.08 | 67.94 | 69.45 | 70.17 |
| wRRF | **35.82** | **58.34** | _65.93_ | **73.25** | 79.76 | **86.29** | **90.01** | **71.34** | **60.15** | **63.99** | **66.90** | **68.94** | **70.56** | **71.30** |

## F  THE USE OF LARGE LANGUAGE MODELS

In this research, Large Language Models (LLMs) tools were utilized to improve both the efficiency of dataset construction and the refinement of the manuscript. During data crawling, GPT-4.1-mini was employed to convert image-based content—such as mathematical formulas and tables—into accessible text format. In the dataset construction phase, GPT-4o was used to refine raw user-submitted questions, transforming them into grammatically correct and complete sentences to ensure clarity and precision. Throughout the writing process, LLMs tools also served as utilities for grammar and spell checking.

## G  EXAMPLES OF KOREA NATIONAL LAW INFORMATION CENTER

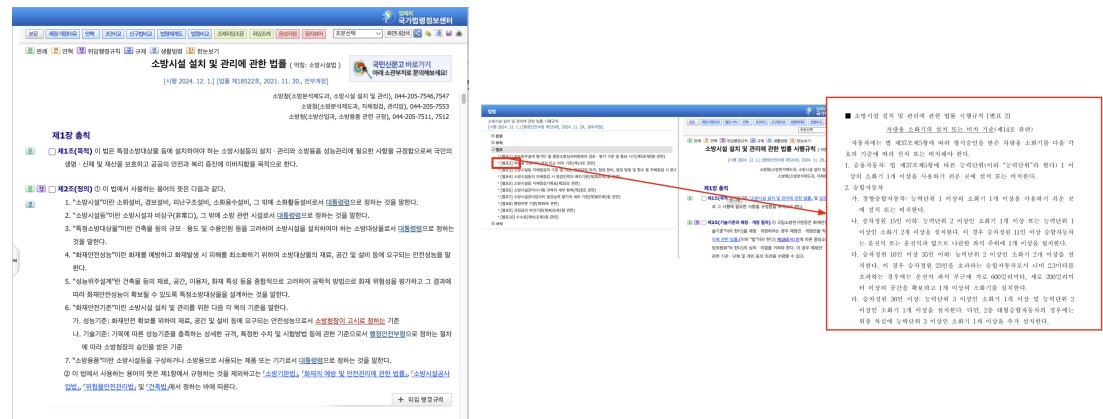

Figure 3: Examples of Korea National Law Information Center

The examples presented in Figure 3 illustrate the structure and content of legal texts retrieved from the Korea National Law Information Center. The left panel displays an excerpt from the *Act on Installation and*

*Management of Firefighting Systems* (Act No. 18522). This section encompasses Chapter 1, detailing the legislative purpose (Article 1) to protect public safety and property through the management of firefighting systems, and the definitions (Article 2) for key terms such as "firefighting systems," "specific fire safety objects," and "performance-based design."

The right panel demonstrates the hierarchical navigation within the *Enforcement Rule* of the same Act, specifically highlighting [Annex 2] titled "Standards for Installation or Placement of Fire Extinguishers for Vehicles." The red arrows serve as a visual guide, tracing the relationship between the appendix directory on the sidebar and the specific regulation text displayed in the main window. This regulation mandates that all vehicles must be equipped with type-approved fire extinguishers. The detailed standards specify that passenger cars must carry at least one extinguisher, while passenger vans are subject to stricter requirements regarding the number and capacity of extinguishers based on their seating capacity (e.g., 15 or fewer, 16–35, and 36 or more passengers).

## H    PROMPTS FOR SYNTHETIC QA GENERATION

This section details the prompts used with GPT-4o to generate the synthetic Single-Hop and Multi-Hop QA datasets.

Table 13: Prompt Template for Multi-Hop QA Generation (Section 4).

```
## Task Instructions
You are tasked with creating a Multiple Choice Question & Answer (MCQA)
↪   set based on the two provided Korean legal documents below. The
↪   primary goal is to design this QA set specifically for evaluating a
↪   Retrieval-Augmented Generation (RAG) system.

### Core Dependency Logic & Constraints
* **Dependency:** The question's answerability must strictly follow the
↪   dependency: **'Document A -> unanswerable; Document A + Document B ->
↪   answerable'**
* **Question Style:** The question must be phrased naturally, without
↪   explicitly citing law or article numbers.
* **Answer:** You can freely set the correct answer number among the
↪   options.

### Required Output Format
1. [Query]
2. [Options]
3. [Answer]
4. [Explanation] (Explaining both the unanswerable and answerable
↪   scenarios)

**Language Instruction:** Your entire response must be **in Korean**.
***
## Provided Context Documents

### Document A:
{document_a}

### Document B:
{document_b}
***
```

## I    PROMPTS FOR OPEN-ENDED QA EVALUATION (LLM-AS-JUDGE)

This section details the prompts used for the LLM-as-Judge metrics (Binary Factuality and Win-Rate) in the Open-Ended QA experiments (Section 5).

Table 14: Prompt Template for Single-Hop QA Generation (Section 4).

```
# INSTRUCTIONS
You are an expert in creating educational quizzes from legal documents.
↪ Your task is to generate **three distinct Yes/No questions** based on
↪ the legal document provided below.

1. Each question must test a key condition or rule from the text.
2. Each question must be answerable with a simple "Yes" or "No".
3. You **must write the entire output in Korean (한국어)**.

Follow this numbered format exactly for each of the three questions:
1. 질문: [Question 1 in Korean]
1. 정답: [Answer 1 in Korean: 예 or 아니오]
1. 해설: [Explanation 1 in Korean]
2. 질문: [Question 2 in Korean]
2. 정답: [Answer 2 in Korean: 예 or 아니오]
2. 해설: [Explanation 2 in Korean]
3. 질문: [Question 3 in Korean]
3. 정답: [Answer 3 in Korean: 예 or 아니오]
3. 해설: [Explanation 3 in Korean]

# LEGAL DOCUMENT
**title: {title}**
content: {document_text}

# GENERATE OUTPUT
```

Table 15: Prompt for LLM-as-Judge (Binary Factuality Evaluation).

**System Prompt:** You are an expert grader. Return ONLY a single character: '1' (if the model answer is factually correct and sufficiently comprehensive relative to the gold answer) or '0' (otherwise). No explanation, no punctuation.
**User Prompt:**

```
### Question
{q}

### Gold Answer
{ref}

### Model Answer
{hyp}

### Task
Judge the model answer. Respond with 1 or 0 only.
```

Table 16: Prompt for LLM-as-Judge (Pairwise Comparison/Win-Rate).

**System Prompt:** You are an expert grader. Reply with a single character: A or B.
**User Prompt:**

```
### Question
{q}

### Relevant Documents
{ctx if ctx else '(None)'}

### Answer A
{A}

### Answer B
{B}

### Task
Assess which answer is **more factually correct and comprehensive** given
↪  the question and the documents.
Reply with *only* `A` or `B`.
```

## J PROMPTS FOR SYNTHETIC QA INFERENCE

This section details the prompts used by the LLMs during inference for the Single-Hop and Multi-Hop QA experiments (Section 5). The original prompts were in Korean and have been translated into English here.

Table 17: Prompts for Multi-Hop QA Inference.

---

**System Prompt (Zero-shot):** You are an evaluator answering the given multiple-choice question. Read the question and options carefully and select the most appropriate answer. Your response must be only the number corresponding to the correct option (e.g., 1, 2, 3, 4, or 5). Do not include any other explanations.

**System Prompt (Context-based: Partial/Full Context):** You are an evaluator answering the multiple-choice question based on the provided context (documents). Your answer must be based solely on the content of the provided context. **Important Instruction:** If the answer to the question cannot be found within the provided context, you must select the option indicating that the information is unknown or cannot be determined (e.g., 'Cannot determine', 'No information'). Your response must be only the number corresponding to the correct option (e.g., 1, 2, 3, 4, or 5). Do not include any other explanations.

**User Prompt Template:**
```
{context_section}
[Question]
{question}

[Options]
{options_text}

[Your Answer (Number only)]
```

---

Table 18: Prompts for Single-Hop QA Inference.

**System Prompt (Zero-shot):** You are an evaluator who answers the given question with only 'Yes' or 'No'. Read the question carefully and respond only with 'Yes' or 'No', without any other explanation.

**System Prompt (Oracle RAG):** You are an evaluator who answers the question based on the provided context (document) with only 'Yes' or 'No'. Your answer must be based solely on the content of the provided context. Respond only with 'Yes' or 'No', without any other explanation.

**User Prompt Template:**

```
{context_section}
[Question]
{question}
[Your Answer ('Yes' or 'No' only)]
```

## K    QUALITATIVE ANALYSIS

While the quantitative evaluation demonstrates a high overall agreement of 88.30% between the LLM-as-a-Judge and human evaluators, an investigation of the remaining discrepancies reveals critical insights into the model's behavior. In this section, we present a qualitative analysis of representative examples corresponding to the four quadrants of the confusion matrix (Table 5). Table 19 details these cases, focusing on the underlying causes of disagreement that metrics alone fail to capture.

Specifically, we examine the raw Korean texts alongside English translations to diagnose distinct failure modes. Despite strong inter-rater agreement ($\kappa = 0.88$) validating the human ground truth, the automated judge exhibited a tendency toward stringency, resulting in a False Negative (FN) rate of 10.80%.

Our analysis highlights two primary error types:

- **Overestimation (False Positive):** Generic answers lacking necessary domain-specific entities (e.g., *Ministry of National Defense*) were frequently rated as valid by the LLM, whereas human evaluators penalized the lack of precision.
- **Underestimation (False Negative):** This error type predominantly occurred when the official reference answers (NFA) were concise. Since the LLM judge treats the official answer as the absolute gold standard, it tended to penalize model responses that provided more specific details than the reference. This structural bias suggests that future iterations may require prompt diversification or an ensemble of evaluator models beyond GPT-4o to mitigate strictness regarding specificity.

Table 19: Qualitative Analysis with Full Text. English translations are provided below each text block.

| Type | Full Comparison (Question / Official / Model) | Rationale |
|---|---|---|
| **TP (True Positive)** **61.20%** | **[Question]**
운동시설 영업장 내부에 바닥에서 반자 내부 천장까지 준불연재료로 구획하고 불연문으로 설치한 복도, 화장실, 계단은 수용 인원 산정 시 바닥 면적에서 제외되는지, 아니면 제외되지 않는지 알 수 있을까요?
In a sports facility business, are corridors, restrooms, and stairs that are partitioned from floor to ceiling with semi-non-combustible materials and installed with non-combustible doors excluded from the floor area calculation when determining occupancy capacity?

**[Official Answer]**
「소방시설 설치 및 관리에 관한 법률 시행령」 별표 7 비고 제1호에 따라 운동시설의 바닥 면적 중 「건축법 시행령」 제2조제11호에 따른 준불연재료 이상의 것을 사용하여 바닥에서 천장까지 벽으로 구획한 복도와 화장실, 계단의 면적은 수용인원 계산시 바닥면적에서 제외할 수 있다고 판단됩니다.
According to Note 1 of Table 7 in the 「Enforcement Decree of the Act on Installation and Management of Firefighting Systems」, it is judged that the area of corridors, restrooms, and stairs in sports facilities, partitioned from floor to ceiling using materials equal to or higher than semi-non-combustible materials pursuant to Article 2, Item 11 of the 「Enforcement Decree of the Building Act」, **can be excluded** from the floor area when calculating occupancy capacity.

**[Model Answer]**
운동시설 영업장 내부에 바닥에서 반자 내부 천장까지 준불연재료로 구획하고 불연문으로 설치한 복도, 화장실, 계단은 수용 인원 산정 시 바닥 면적에서 제외됩니다. 이는 소방시설 설치 및 관리에 관한 법률 시행령 별표 7의 비고 1에 따라, 준불연재료 이상의 것을 사용하여 바닥에서 천장까지 벽으로 구획한 복도, 계단 및 화장실의 바닥면적은 포함하지 않기 때문입니다.
Corridors, restrooms, and stairs inside a sports facility business that are partitioned from floor to ceiling with semi-non-combustible materials and installed with non-combustible doors **are excluded** from the floor area calculation when determining occupancy capacity. This is because, according to Note 1 of Table 7 of the Enforcement Decree of the Act on Installation and Management of Firefighting Systems, the floor area of corridors, stairs, and restrooms partitioned from floor to ceiling using materials equal to or higher than semi-non-combustible materials **is not included.** | **Accurate Legal Reasoning:** The model correctly identified that the area is excluded. It accurately cited the relevant regulation ("Enforcement Decree Table 7, Note 1") and matched the official interpretation perfectly. Both the LLM-judge and human evaluator marked this as correct. |

**Table 19 – continued from previous page**

| Type | Full Comparison (Question / Official / Model) | Rationale |
|---|---|---|
| **FP (False Positive)** 0.90% | **[Question]**
건축물의 건축 행위로 소방시설을 설치할 경우, 소방시설법 제22조 제2항 제1호에 따라 건축물 사용승인일로부터 60일 이내에 최초 점검을 해야 합니다. 그러나 군부대의 경우, 여러 사유로 인해 건축물대장에 사용승인일이 기재되어 있지 않거나 사용승인일을 알 수 없는 경우에는 최초 점검을 언제 해야 하나요?
When installing firefighting systems due to construction acts, an initial inspection must be conducted within 60 days of the usage approval date pursuant to Article 22, Paragraph 2, Item 1 of the Firefighting Systems Act. However, in the case of military units, if the usage approval date is not recorded in the building ledger or is unknown for various reasons, when should the initial inspection be conducted?

**[Official Answer]**
최초점검은「소방시설 설치 및 관리에 관한 법률」제22조 제1항 제 1호에서「건축법」제22조에 따라 건축물을 사용할 수 있게 된 날부터 60일 이내 최초점검을 받도록 규정하고 있습니다. - 군부대의 경우 건축물 사용승인 관련 일자를 국방부에 문의하여 확인하시기 바랍니다.
The initial inspection is required to be conducted within 60 days from the date the building becomes available for use pursuant to Article 22 of the「Building Act」, as stipulated in Article 22, Paragraph 1, Item 1 of the「Act on Installation and Management of Firefighting Systems」. - In the case of military units, please inquire with the Ministry of National Defense to confirm the dates related to building usage approval.

**[Model Answer]**
군부대의 경우 ... (중략) ... 구체적인 점검 시기는 행정안전부령에 따르므로, 관련 규정을 확인하거나 소방본부장 또는 소방서장에게 문의하여 지침을 받는 것이 필요합니다. ... (후략)
In the case of military units, if the usage approval date is unknown... (omitted) ... As the specific inspection timing follows the Ordinance of the Ministry of the Interior and Safety, it is necessary to check relevant regulations or inquire with the Fire Headquarters or Fire Station Chief for guidance. ... (omitted) | **Judge Overestimation:**
The model correctly identified a legislative silence regarding specific inspection timelines for military units lacking usage approval dates. Given that no explicit provision exists in the standard Firefighting Systems Act for this edge case, the model's recommendation to **inquire with administrative authorities** represents a procedurally sound solution.
However, the LLM Judge overestimated the significance of the lexical deviation. By fixating on the absence of the specific keyword "Ministry of National Defense," the evaluator failed to recognize that the model's advice is the functionally equivalent and correct course of action in the absence of codified regulations. |

**Table 19 – continued from previous page**

| Type | Full Comparison (Question / Official / Model) | Rationale |
|------|-----------------------------------------------|-----------|
| **FN (False Negative)** **10.80%** | **[Question]**
외기 개방구조 차량 램프 구간에 감지기를 설치하지 않아도 되는 장소가 있나요?
Are there any locations within vehicle ramp sections with structures open to outside air where detectors do not need to be installed?

**[Official Answer]**
해당 장소는 「자동화재탐지설비 및 시각경보장치의 화재안전기술기준(NFTC 203)」 2.4.5.2 '헛간 등 외부와 기류가 통하는 장소로서 감지기에 따라 화재 발생을 유효하게 감지할 수 없는 장소'에 해당할 수 있을 것으로 판단되며, 감지기 제외에 대한 세부적인 사항은 관할소방서와 협의하시기 바랍니다.
It is judged that the location may fall under Section 2.4.5.2 of the 「National Fire Safety Technical Code for Automatic Fire Detection Systems and Visual Alarm Devices (NFTC 203)」, which refers to 'places such as barns where air flows with the outside, making it impossible for detectors to validly detect fire.' Details regarding detector exclusion should be discussed with the competent fire station.

**[Model Answer]**
외기 개방구조 차량 램프 구간에 감지기를 설치하지 않아도 되는 장소는 다음과 같습니다: 1. 천장 또는 반자의 높이가 20 m 이상인 장소 ... 2. 헛간 등 외부와 기류가 통하는 장소로서 ... 3. 부식성 가스가 체류하고 있는 장소. 4. 고온도 및 저온도로서 ... 5. 목욕실, 욕조나 샤워시설이 있는 화장실 ... 6. 파이프덕트 등 그 밖의 이와 비슷한 것 ... 7. 먼지, 가루 또는 수증기가 다량으로 체류하는 장소 ... 8. 프레스공장, 주조공장 등 화재 발생의 위험이 적은 장소 ... 이러한 장소들은 감지기를 설치하지 않아도 되는 예외적인 경우에 해당합니다.
Locations ... where detectors do not need to be installed are as follows: 1. Places with ceiling height of 20m or more ... 2. Places like barns where air flows with the outside ... 3. Places with corrosive gas. 4. High or low temperature places ... (Items 5-8 omitted for brevity in translation, but present in Korean text) ... These locations correspond to exceptional cases. | **Judge Underestimation:** The official answer is cautious ("consult the fire station"), whereas the model provides the full technical list of exemptions from the relevant standard (NFTC 203). The Judge likely penalized the model for providing a list instead of the specific advice to "consult," while the human evaluator recognized the correctness and utility of the list. |

**Table 19 – continued from previous page**

| Type | Full Comparison (Question / Official / Model) | Rationale |
|------|-----------------------------------------------|-----------|
| **TN (True Negative)** 27.10% | **[Question]**
전기저장장치 설비의 송수구는 다른 설비의 송수구와 별도로 구분하여 설치해야 하나요, 아니면 스프링클러 송수구와 함께 사용해도 되나요?
Should the water inlet for the Energy Storage System (ESS) be installed separately from the inlets of other systems, or can it be used together with the sprinkler inlet?

**[Official Answer]**
「전기저장시설의 화재안전기술기준(NFTC 607)」2.2.1.7에 따라 전기저장장치 설비에 송수할 수 있는 송수구를 별도로 설치해야 한다고 판단됩니다.
According to Section 2.2.1.7 of the 「National Fire Safety Technical Code for Energy Storage Facilities (NFTC 607)」, it is judged that a water inlet capable of supplying water to the ESS **must be installed separately.**

**[Model Answer]**
전기저장장치 설비의 송수구는 스프링클러 설비의 화재안전기술기준(NFTC 103) 2.8에 따라 설치해야 합니다. 따라서 전기저장장치 설비의 송수구는 스프링클러 설비의 송수구와 함께 사용할 수 있습니다. 별도로 구분하여 설치할 필요는 없습니다.
The water inlet for the ESS must be installed in accordance with NFTC 103 2.8. Therefore, the ESS water inlet **can be used together** with the sprinkler inlet. It is not necessary to install it separately. | **Detection of Hallucination:** The model incorrectly cited NFTC 103 (Sprinklers) instead of the specific ESS standard (NFTC 607), leading to the wrong conclusion (shared vs. separate). The Judge correctly identified this factual contradiction. |