# OpenReview forum: "SearchFireSafety: A Retrieval-Augmented Legal QA Dataset for Fire Safety"
_ICLR.cc/2026/Conference — ICLR 2026 Conference Withdrawn Submission_

### Official Review · Reviewer_Sr1D · 2025-10-20

**Soundness:** 3
**Presentation:** 2
**Contribution:** 2
**Rating:** 4
**Confidence:** 4

**Summary:**

The paper introdcues three datasets around South Korean fire safety regulations: one real-world dataset scraped from actual queries, and two synthetic ones (once MCQA with multihop across two documents, once binary with one document). The authors show across many experiments that these tasks are hard for models, and that legal retrieval is still challenging.

**Strengths:**

- There's a lot to be liked about this paper. I am deeply interested in legal retreival, and would instantly believe all the findings the authors made.
- The real-world dataset is very interesting, and I think that's a useful resource.

**Weaknesses:**

- There's an incoherent narrative and a bit too much cramped into one paper: Is the main contribution the three datasets together, the retrieval part, the generation part, the hallucination part? Because of that, the individual contributions seem to be a bit shaky and not as coherent (this is the main reason for my clarity score, the paper was easy to read and well written.)
- Why don't the authors use the synthetic datasets for training? This would be the big contribution: Synthetic data helps real-world legal retrieval or generation (for generation, the datasets probably would have to be constructed differently). If it doesn't help for training, why is that, why doesn't synthetic data work? What would work instead?
- Perhaps I'm overcritical of the synthetic datasets, but as is, only used to evaluate models, I don't see their utility and think the paper would be stronger without them.
- Related to this, the authors seem to have done lots of manual annotation work which is highly appreciated. While they give qualitative examples of why certain things are difficult for models, an extensive manual error analysis would be greatly appreciated, and could inform future research on legal retrieval and highlight what currently doesn't work and why.
- Having said all of that, again I think there's a lot here. I wouldn't mind seeing this paper accepted, but I would be a bit sad because right now it seems to be a somewhat incoherent contribution, and easily could be a very compelling contribution with not that much more work.

**Questions:**

- line 45:  "crucial yet under-examined domain" -- I think this is overstating things slightly. How about "important yet ..."
- Line 142: Perhaps cite https://dl.acm.org/doi/pdf/10.1145/3709025.3712219 here? They discuss similar topics
- Would be curious to see how Qwen3 embeddings and rerankers perform on this dataset: https://huggingface.co/Qwen/Qwen3-Embedding-8B
- section 5.2.2 and 5.2.1 seem to be somewhat contradictory. If there's only limited lexical overlap between documents and answers, how can ROUGE scores improve in a RAG setting? Would like to see more details on this
- Table 6 vs Table 8: This also seems curious: table 8 metrics are way higher. Now, I wouldn't expect the tasks to be way harder given full context. It reminded me of https://arxiv.org/abs/2505.12864 though, where the authors added multiple choice options and model performance went down a lot, perhaps try something similar here as well?

---

> ### Author Response · Authors · 2025-11-24
>
> ## Response to Reviewer Sr1D
>
> **W1: Incoherent narrative and too much cramped into one paper.**
>
> We sincerely appreciate this thoughtful and constructive feedback. The initial version indeed attempted to cover too many components at once, which made the narrative feel fragmented. Following the reviewer’s guidance, we undertook a substantial revision of the paper.
>
> In the revised manuscript, we reorganized the entire narrative so that SearchFireSafety is presented not as a set of loosely connected tasks, but as a coherent and comprehensive diagnostic suite. We also rewrote the introduction and discussion sections to clearly highlight this unified perspective.
>
> For further details, please refer to Response to Common Concerns (R1-4).
>
> ---
> **W2 & W3: Utility of synthetic datasets and why not used for training.**
>
> Please refer to R4. The synthetic datasets provide controlled environments to diagnose specific failure modes, particularly uncertainty awareness (Table 7). Our focus was rigorous benchmarking, not training.
>
> ---
>
> **W4: Extensive manual error analysis would be appreciated.**
>
> We agree that detailed manual error analysis is essential for validating the reliability of the LLM-as-a-Judge metric. While the main paper reported only the aggregate agreement, we now include a fuller analysis in Appendix K (Table 19).
>
> Although the LLM judge aligns with human evaluators in 88.30% of cases, the remaining discrepancies provide important insight. Our results show that most disagreements stem from the judge being overly strict, resulting in a 10.80% false-negative rate.
>
> The key error types are:
> - False Positives: The LLM sometimes accepts generic answers missing critical domain-specific entities, whereas human evaluators require this precision.
> - False Negatives: When official reference answers are brief, the LLM penalizes more detailed—but still correct—responses. This reflects an over-literal interpretation of the gold answer.
>
> These findings clarify the judge’s behavior and reinforce the need for more diverse prompts or evaluator ensembles in future work.
>
> ---
>
> **Q1: Line 45: "crucial yet...".**
>
> We agree and revise it to "under-examined but socially important domain" (Line 43).
>
> ---
>
> **Q2: Line 142: Citation suggestion.**
>
> Thank you. We will include the suggested citation.
>
> ---
>
> **Q3: Qwen3 embeddings performance.**
>
> As noted in Table 3 (revised version: Table 10), Qwen3-Embedding-0.6B was included in our evaluation. It performed strongly (MRR 31.55), second only to BGE-m3, further supporting our finding that multilingual models are well-suited for this domain.
>
> Although Qwen3-8B was excluded from our primary pipeline due to inference latency constraints, we conducted additional experiments in response to the reviewer’s suggestion.
>
> Qwen3-8B achieved an MRR of 83.91 (vs. 31.55) and a Recall@10 of 77.07 (vs. 48.23). These results demonstrate that, despite increased computational cost, scaling the embedding model provides substantial gains in retrieval accuracy.
>
> ---
>
> **Q4: Section 5.2.2 and 5.2.1 contradiction (Lexical overlap vs. ROUGE).**
>
> This is an insightful observation. While queries written by non-experts have limited lexical overlap with documents (hindering sparse retrieval, Sec 5.2.1), the gold answers (written by experts) naturally use formal legal terminology found in the documents. Therefore, when RAG provides the correct context, the generated answer incorporates this terminology, increasing ROUGE with the gold answer (Sec 5.2.2).
>
> ---
>
> **Q5: Table 6 vs Table 8 (Multi-hop vs Single-hop performance).**
>
> The gap highlights the inherent difficulty of multi-hop legal reasoning. Table 8 (Single-hop) scores are much higher (up to 96.50%) than Table 6 (Multi-hop) (up to 77.31%) in the Full Context setting. Synthesizing information across two interconnected documents is significantly more challenging than extracting information from a single document, even when all context is provided.

---

### Official Review · Reviewer_KC4s · 2025-10-30

**Soundness:** 2
**Presentation:** 2
**Contribution:** 3
**Rating:** 2
**Confidence:** 2

**Summary:**

This paper proposes a new benchmark, SEARCHFIRESAFETY, for the South Korean fire safety regulation domain, designed to evaluate the RAG capabilities of models. The dataset includes 941 real-world, open-ended QA pairs and over 4,000 synthetic QA pairs. The paper also provides a corpus of 4,437 legal documents from 117 statutes, complete with a citation graph, to serve as the source for RAG. Experiments demonstrate that multilingual dense retrieval performs best on this benchmark, but LLMs with RAG still struggle with multi-hop legal QAs.

**Strengths:**

1. The paper addresses the RAG problem within the South Korean fire safety regulation domain, which is a novel and underexplored area.

2. The authors conduct comprehensive experiments, comparing a wide variety of retrieval methods and the capabilities of several different LLMs.

**Weaknesses:**

1. My major concern is the dataset's quality, which appears highly dependent on the authors, who also served as the annotators. The paper lacks a sufficient qualitative or quantitative analysis of the dataset. Simply stating that the data was human-reviewed is not enough; a more rigorous methodology is needed to validate the dataset's quality. Specifically:

* Real-World Open-Ended QA: The annotation procedure does not seem reliable, making the benchmark's quality questionable. As stated in lines 137-140, the dataset was annotated by the authors. However, it is not specified whether the authors possess the necessary legal expertise to annotate such questions. Furthermore, the Inter-Annotator Agreement (IAA) is missing, which casts doubt on the reliability and consistency of the annotations. Additionally, it appears the annotators only validated the documents retrieved by BM25 rather than identifying all possible relevant documents. Given that BM25 cannot guarantee 100% recall, the final dataset likely suffers from missing relevant documents (low recall).

* Legal Document Corpus: Coverage is a significant issue. According to lines 211-212, the corpus was collected by crawling only those sources cited at least twice in the NFA answer set and their parent instruments. Any document not meeting this criterion is excluded. A legal document dataset should aim for comprehensive coverage. Limiting the corpus only to documents relevant to the NFA answer set greatly diminishes its usability and generalizability.

* Synthetic QA Dataset: The generation method is ambiguously described. The authors state they "use the citation graph," but this is too vague. More details are necessary. Besides, the quality of the final synthetic dataset is unknown, apart from the authors' claim to have "checked them."

2. The paper's focus is excessively narrow. While South Korean fire safety regulation is a valuable domain, the paper would be much stronger if it included at least one other domain. The described construction pipeline for both the QA dataset and the legal corpus appears to be highly tailored to this specific domain, which raises concerns about the generalizability of the proposed methods.

3. The second stated challenge of RAG in the legal domain is not clearly articulated. In lines 070-073, the authors define this challenge as: "legal documents are interconnected through a dense web of statutory cross-references... many of which are vague or overly broad." It is not immediately clear why this interconnectedness and vagueness inherently make RAG significantly harder. The paper needs to elaborate on this point to make the challenge concrete.

**Questions:**

Please refer to the weaknesses.

---

> ### Author Response · Authors · 2025-11-24
>
> ## Response to Reviewer KC4s
>
> **W1: Concerns about dataset quality (Annotation, Coverage).**
>
> Please refer to our detailed response in R1 regarding annotation expertise (NFA experts provided answers), IAA (High Cohen's Kappa), BM25 usage (initial candidates only), and corpus coverage (principled scope).
>
> ---
>
> **(W1c) Synthetic QA Dataset generation method.**
>
> We apologize for any lack of clarity in the original description. We have revised Section 4 accordingly and added an MCQA example (Table 3) to make the process more transparent.
>
> Our methodology is grounded in the structure of the citation graph and follows a strict dependency rule: each question must be unanswerable using Document A alone but become answerable when combined with its referenced Document B. GPT-4o was used for initial question generation, after which all items underwent rigorous human validation. Importantly, 21% of the generated items (1,076 in total) were discarded for failing to satisfy this dependency requirement, ensuring the robustness and reliability of the final dataset.
>
> ---
>
> **W2: The paper's focus is excessively narrow.**
>
> Please refer to R2 and R3.
>
> ---
>
> **W3: The second stated challenge (interconnectedness and vagueness) is not clearly articulated.**
>
> Please refer to R2.

---

### Official Review · Reviewer_S1E4 · 2025-11-01

**Soundness:** 3
**Presentation:** 3
**Contribution:** 2
**Rating:** 2
**Confidence:** 4

**Summary:**

This paper targets the problem of a lack of a QA dataset for the domain of South Korean fire safety regulation. It proposes a dataset of real-world and synthetic question-answers with 941 real-world open-ended QA and 4437 legal documents. The legal document dataset was constructed with a web crawling process, a human-in-the-loop process for verifying legal documents, and a citation graph construction representing the relevant between documents. The experiments evaluate the quality of the RAG-based model built on the proposed dataset in both the retrieval and generation phases. For retrieval, Recall@k is the metric used for evaluation. It shows that weighted Reciprocal Rank Fusion (wRRF) outperformed other techniques in this domain over the real-world open-ended QA. For generations, the ROUGE and BERTScore were used as the textual matching score, while LLM-AS-A-Judge was used with the integration of a strong close LLM for judgment, along with the Win-Rate score. Accuracy shows that the best context retrieved for answer generation for each question is the Gold context (i.e., the corresponding legal document that contained the question). Authors use the synthetic dataset to evaluate the performance of the RAG approach over different LLMs. It shows that, interestingly, with Full Context, an open LLM with 8B parameters can outperform the well-known GPT-4o model in getting more correct answers.

**Strengths:**

- The dataset construction process (both questions and legal documents) was performed on reliable resources.
- While collecting real real-world dataset of questions requires significant cost for humans as experts to evaluation questions, the second strategy for question creation, which focuses on yes/no questions and multiple-choice questions, requires significantly less cost and is applicable to other domains.

**Weaknesses:**

- Overall, while this work highlights important observations about the roles of the retrieval process in achieving good accuracy, it lacks contributions regarding the improvement of RAG models.
- In the introduction, the authors claimed that the legal framework governing fire safety in South Korea is complex and fragmented, which doesn’t convince me. Authors should provide clarifications about the reasons for this challenge. Additionally, I don’t see any significant challenges in building a RAG-based dataset for the Korean language compared to other languages.
- A minor point, figures should be translated into English to ensure they are understandable to reviewers.

**Questions:**

- The metric used for Generation can be extended if authors use different models (i.e., Claude, Gemini), replacing GPT-4o as LLM-As-a-judge.
- How many experts are assigned to validate a given question/ legal document? How do you solve the conflict between experts’ opinions?
- What are the distinctions between constructing the South Korean Fire Safety dataset compared to other language datasets?

---

> ### Author Response · Authors · 2025-11-24
>
> ## Response to Reviewer S1E4
>
> **W1: Lacks contributions regarding the improvement of RAG models.**
>
> Please refer to R4. While our focus is the benchmark, we introduced wRRF, a hybrid retrieval strategy tailored to the query distribution of this domain, which yielded the best retrieval performance (Table 10).
>
> ---
>
> **W2: Clarification on the challenges of the South Korean fire safety framework.**
>
> Please refer to R2.
>
> ---
>
> **W3: Figures should be translated into English.**
>
> We apologize for this oversight (e.g., Figure 3 in Appendix G). We have translated the content into English in the revised version.
>
> ---
>
> **Q1: Extending LLM-As-A-Judge with other models (Claude, Gemini).**
>
> We chose GPT-4o due to its strong performance and high correlation with human judgment in our analysis (88.30% agreement, Table 5). Given this high reliability, we believe it serves as a robust standard for evaluation.
>
> ---
>
> **Q2: Number of experts and conflict resolution.**
>
> Please refer to R1. Gold answers in real-world QA were written by NFA experts.
>
> ---
>
> **Q3: Distinctions of constructing the South Korean Fire Safety dataset.**
>
> Please refer to R2.

---

### Official Review · Reviewer_VTNK · 2025-11-01

**Soundness:** 3
**Presentation:** 3
**Contribution:** 3
**Rating:** 6
**Confidence:** 3

**Summary:**

This paper develops a retrieval-augmented legal question answering dataset on South Korean fire safety regulation. It contains 941 real-world QA pairs and also synthetic a MCQA corpus targeted at evaluating LLM hallucination on legal reasoning.

**Strengths:**

I think this paper presents a solid, clear and verifiable evaluation of legal RAG. It is great that it is in QA format, which allows robust ground truth, and even using LLM-as-a-judge, we can see there's a high correlation with existing reference-based metrics. There are also plenty of ablations/strategies of your method. The evaluation feels intuitive and objective. I would imagine criticisms at the niche domain of this paper, but I think that's less of a concern as this is a neat dataset which can measure general legal reasoning abilities. I also like that the paper also constructs a synthetic QA corpus to evaluate LLM hallucination with partial context, which is very thoughtful.

**Weaknesses:**

The main problem I have is that for evaluating open-ended QA, even though I think it makes sense that reference-based metrics might already be robust to reflect the answers to open-ended questions (e.g. from https://arxiv.org/abs/2305.12421) It still might be useful to show some qualitative examples and make a more convincing case that your open-ended eval is reliable.

**Questions:**

What do you think can help LLMs improve on this benchmark? Does scaling help (and if so, what kind of data) or is there something more that should be involved? This is irrelevant of my rating but just curious about how the authors think and totally optional to answer.

---

> ### Author Response · Authors · 2025-11-24
>
> ## Response to Reviewer VTNK
>
> **W1: Reliability of open-ended QA evaluation.**
>
> We appreciate this suggestion. As shown in Table 5, our LLM-as-a-Judge (using GPT-4o) achieves 88.30% agreement with human annotations. We observed that the LLM judge is often stricter than human experts (10.80% False Negatives). To make a more convincing case, we have included an appendix with qualitative examples demonstrating cases where ROUGE/BERTScore are high but the answer is factually incorrect (e.g., the "tamper switch" example, L341-351).
>
> ---
>
> **Q1: How to improve LLMs on this benchmark? (Optional)**
>
> We investigated this by conducting **Continual Pre-Training (CPT)** on Qwen3-8B using a legal corpus from AIHub during the rebuttal. The results were illuminating:
> 1.  **Performance Jump:** CPT drastically improved performance, achieving **58.89** (Zero-Shot) and **78.68** (Full Context). This effectively matches GPT-4o level performance using only an 8B model, demonstrating the effectiveness of **domain knowledge-aware training**.
> 2.  **The Trade-off (Validation of Benchmark):** However, in the **Partial Context** scenario, performance dropped to **39.44**. This suggests that while CPT injected knowledge, it also increased the model's tendency to hallucinate answers when context is insufficient, rather than abstaining.
> This experiment highlights the value of our benchmark: without the **Partial Context** metric, one might falsely conclude that CPT is a "silver bullet." Our benchmark provides a nuanced view, balancing knowledge acquisition against hallucination risks. We have added these findings to the revision to strengthen the analysis of model improvements (Table 9).

---

> > ### Comment · Reviewer_VTNK · 2025-11-26
> > **Thank you for your response**
> >
> > Dear authors,
> > Thank you for your response. Since I have already given the highest score among the reviewers, I won't raise scores but maintain my positive assessment. My opinion of this paper is that it's good science and has relatively high result validity. I can see why other reviewers may think the domain is too narrow - but I think this really depends on each person's perspective. Best of luck with submission and rebuttal!

---

### Author Response · Authors · 2025-11-24
**Response to Common Concerns [1/2]**

Dear Reviewers,

We sincerely thank you for your thoughtful and constructive feedback on our submission, “SearchFireSafety: A Retrieval-Augmented Legal QA Dataset for Fire Safety.” We appreciate the reviewers’ recognition that our work offers a “solid, clear, and verifiable evaluation of legal RAG” (VTNK), addresses a “novel and underexplored area” (KC4s), provides a “useful real-world dataset” (Sr1D), and relies on “reliable resources” during dataset construction (S1E4).

We have carefully considered all comments and offer detailed responses below. We begin by addressing the major recurring concerns regarding dataset quality, domain complexity, and contribution clarity.

---

## Response to Common Concerns
### R1. Dataset Quality and Annotation Reliability (KC4s-W1, S1E4-Q2)

We appreciate the opportunity to clarify our annotation process and evaluation methodology.

- **Expertise and Gold Standard (KC4s-W1a):** All gold-standard answers originate from official responses authored by officers of the Korean National Fire Agency (NFA). These officers explicitly cite the relevant legal provisions in their explanations. Importantly, the authors did not generate these answers; our role was limited to dataset curation and verification of document–answer mappings. These details are now included in the revised manuscript (Lines 118–122).

- **Inter-Annotator Agreement & Evaluation Reliability (KC4s-W1a, S1E4-Q2)**: To validate the reliability of our LLM-as-a-Judge metric, two authors independently evaluated model outputs against the NFA gold standard and the referenced legal texts. This yielded a strong inter-rater agreement (Cohen’s κ = 0.88), demonstrating that the provided legal context enables consistent evaluation without requiring external legal counsel. These results are reported in the revised manuscript (Lines 325–339).

- **Corpus Coverage (KC4s-W1b):** Our goal is to define a principled and practically relevant scope aligned with the real-world inquiries collected from NFA. Rather than pursuing an exhaustive legal corpus, we deliberately focused on statutes that are directly relevant for evaluating RAG in practical fire safety scenarios. We have expanded this explanation in the revised manuscript (Lines 204–213).

---

### R2. Domain Complexity and Specificity (S1E4-W2/Q3, KC4s-W3)

We argue that the Korean Fire Safety domain serves as a meaningful representative stress test for legal RAG due to its structural, linguistic, and temporal challenges.

1. **Structural Complexity (KC4s-W3, S1E4-W2):** The domain’s multi-layered structure presents real challenges for retrieval-augmented systems:

   * **Regulatory Fragmentation Across Statutes:** Fire safety requirements are not encapsulated in a single “Fire Act” [1]. They are dispersed across the Building Act, Welfare of the Aged Act, School Facilities Act, and more. Our dataset faithfully reflects this cross-domain dependency, requiring models to synthesize distributed legal information.

    * **Hierarchical Depth & Multi-Hop Reasoning:** Korean law follows a hierarchical structure (Act → Decree → Rule → Administrative Rule). Many provisions contain abstract delegations (e.g., “as prescribed by Presidential Decree”), forcing RAG systems to retrieve and resolve implicit references across multiple levels.

    * **Temporal Dynamics & Hallucination Risk (2022 NFSC Reform):** The 2022 split of the National Fire Safety Code (NFSC) into NFPC and NFTC frequently triggers hallucinations in standard LLMs and search engines, which often cite obsolete NFSC clauses due to parametric memory or outdated indexing. This highlights the need for RAG grounded in curated, temporally accurate sources.

2. **Linguistic Challenges (S1E4-Q3)**: Korean legal texts intermix modern Korean, English loanwords, and specialized Sino-Korean terminology. As shown in Appendix C, this linguistic mixture significantly favors multilingual dense retrievers over monolingual models, underscoring the need for tailored retrieval strategies.

---

> ### Author Response · Authors · 2025-11-24
> **Response to Common Concerns [2/2]**
>
> ### R3. Addressing the “Narrow Focus” Concern (KC4s-W2):
>
> Although our domain is specialized, it is not unusually narrow when viewed in the context of contemporary benchmark development. In fact, recent progress in RAG and legal/technical QA increasingly depends on domain-specific evaluations to meaningfully measure LLM performance. The challenges embedded in our dataset—fragmented citation structures, semantic gaps, and temporal versioning—reflect the realities of regulatory environments that require precise retrieval and robust reasoning.
>
> * **Domain-Specific Benchmarks Are Becoming Central to LLM Evaluation:** Recent benchmarks such as τ-bench [2] (retail/airline operations) and AA-Omniscience [3] (contract law, criminal law, and other specialized domains) explicitly target vertical, domain-grounded settings. These works demonstrate that domain specificity is now recognized as a strength, not a limitation.
>
> * **Korean Benchmarks Follow the Same Trend:** Korean datasets such as ixi-GEN [4] focus on telecom-related industrial consultations. Although these differ from statutory interpretation, they similarly embrace domain specialization to evaluate models under realistic, domain-grounded constraints.
>
> * **Our Dataset Fits This Emerging Paradigm:** In line with these efforts, SearchFireSafety offers a domain where legal reasoning requires integrating cross-statute references, resolving multi-level legal hierarchies, and handling temporally shifting norms (e.g., the 2022 split of the Korean administrative code from NFSC into NFPC and NFTC). The technical specificity of the domain also ensures minimal data contamination in pretraining corpora, enabling our benchmark to evaluate true reasoning rather than memorized knowledge.
>
> In summary, domain specificity is not a weakness of our work but a necessary characteristic of modern RAG benchmarks that aim to expose LLM limitations in complex, high-stakes settings.
>
>
> ---
>
>
>
> ### R4. Clarifying Contributions (Sr1D-W1/W2/W3, S1E4-W1)
>
> We thank the reviewers for highlighting opportunities to improve clarity. We have revised the manuscript accordingly.
>
> 1. **Explicit Contribution Statement (Sr1D-W1):** We have restructured the Introduction to clearly enumerate our contributions, added concrete examples of synthetic data generation (Table 3), and inserted a dedicated explanation of our evaluation protocol (Lines 269–281). Terminology has been standardized across Tables 4 and 6.
>
> 2. **Effect of Training on Legal-Domain Corpus (Sr1D-W1, VTNK-Q1):** During rebuttal, we conducted additional experiments to evaluate the effect of Continued Pretraining (CPT) on a legal-domain corpus. As shown in revised Table 9, CPT substantially improves standard accuracy (approaching GPT-4o), but harms performance in Partial Context settings (dropping to 39.44%). This reveals a crucial trade-off: while CPT enhances factual recall, it also increases hallucination tendencies. Our multi-faceted benchmark is therefore essential for detecting the balance between accuracy and uncertainty awareness—a nuance that simple QA metrics would miss.
>
> 3. **RAG Improvement via Weighted RRF (S1E4-W1):** Beyond dataset creation, our analysis provides methodological insights. We introduced a weighted Reciprocal Rank Fusion (wRRF) approach tailored to legal retrieval (Appendix D). Based on query distribution analysis (noting that ~15% of queries explicitly reference statutes), we optimized wRRF with a 1:9 sparse–dense weighting, achieving the best overall retrieval performance (Table 10).
>
> 4. **Utility of Synthetic Data (Sr1D-W2/W3):** Our synthetic datasets serve a diagnostic role, providing controlled environments for evaluating capabilities that are difficult to isolate in real-world settings—particularly uncertainty awareness. As shown in the Partial Context experiments (Table 7), these datasets are critical for rigorous benchmarking. While future work may explore their use in training (e.g., SFT), their primary function here is evaluation.
>
> ---
>
> ### Reference
>
>
> [1] Ji-hyeok Song. Research on methods to enhance building fire safety via amendments to the fire services
> act. Master’s thesis, Sungkyunkwan University, Seoul, South Korea, February 2023. URL https://www.riss.kr/link?id=T16647456.
>
>
> [2] Yao, S., Shinn, N., Razavi, P., & Narasimhan, K. (2024). $\tau $-bench: A Benchmark for Tool-Agent-User Interaction in Real-World Domains. arXiv preprint arXiv:2406.12045.
>
> [3] Jackson, D., Keating, W., Cameron, G., & Hill-Smith, M. (2025). AA-Omniscience: Evaluating Cross-Domain Knowledge Reliability in Large Language Models. arXiv preprint arXiv:2511.13029.
>
> [4] Kim, S., Na, Y., Kim, K., Cho, H., Lim, G., Kim, M., ... & Jeon, B. K. (2025, November). ixi-GEN: Efficient Industrial sLLMs through Domain Adaptive Continual Pretraining. In Proceedings of the 2025 Conference on Empirical Methods in Natural Language Processing: Industry Track (pp. 2387-2404).

---

### Note · Authors · 2025-12-16

**Comment:**

We have decided to withdraw our submission. Unfortunately, unexpected technical instability on the OpenReview platform during the rebuttal period hindered our ability to fully address the reviewers' concerns and proceed with the discussion.

While we regret missing the opportunity to clarify our work under these circumstances, we sincerely thank the reviewers and ACs for their time and feedback. We will refine our paper for a future submission.

**Withdrawal Confirmation:**

I have read and agree with the venue's withdrawal policy on behalf of myself and my co-authors.